::: PLOS ONE

# Non-gradient and genotype-dependent patterns of RSV gene expression

**Felipe-Andrés Piedra**[1]*, **Xueting Qiu**[2], **Michael N. Teng**[3], **Vasanthi Avadhanula**[1], **Annette A. Machado**[1], **Do-Kyun Kim**[4], **James Hixson**[4], **Justin Bahl**[2,5], **Pedro A. Piedra**[1,6]

**1** Department of Molecular Virology & Microbiology, Baylor College of Medicine, Houston, TX, United States of America, **2** Center for the Ecology of Infectious Diseases, Department of Infectious Diseases, College of Veterinary Medicine, University of Georgia, Athens, GA, United States of America, **3** Division of Allergy and Immunology, Department of Internal Medicine, University of South Florida Morsani College of Medicine, Tampa, FL, United States of America, **4** Human Genetics Center, School of Public Health, University of Texas Health Science Center, Houston, TX, United States of America, **5** Program in Emerging Infectious Diseases, Duke-National University of Singapore Graduate Medical School, Singapore, **6** Department of Pediatrics, Baylor College of Medicine, Houston, TX, United States of America

* Felipe-Andres.Piedra@bcm.edu

**Data Availability Statement:** Sequences reported in this study were deposited in GenBank database under accession numbers MG813977-MG813995.

**Funding:** The author(s) received no specific funding for this work.

## Abstract

Respiratory syncytial virus (RSV) is a nonsegmented negative-strand RNA virus (NSV) and a leading cause of severe lower respiratory tract illness in infants and the elderly. Transcription of the ten RSV genes proceeds sequentially from the 3' promoter and requires conserved gene start (GS) and gene end (GE) signals. Previous studies using the prototypical GA1 genotype Long and A2 strains have indicated a gradient of gene transcription extending across the genome, with the highest level of mRNA coming from the most promoter-proximal gene, the first nonstructural (NS1) gene, and mRNA levels from subsequent genes dropping until reaching a minimum at the most promoter-distal gene, the polymerase (L) gene. However, recent reports show non-gradient levels of mRNA, with higher than expected levels from the attachment (G) gene. It is unknown to what extent different transcript stabilities might shape measured mRNA levels. It is also unclear whether patterns of RSV gene expression vary, or show strain- or genotype-dependence. To address this, mRNA abundances from five RSV genes were measured by quantitative real-time PCR (qPCR) in three cell lines and in cotton rats infected with RSV isolates belonging to four genotypes (GA1, ON, GB1, BA). Relative mRNA levels reached steady-state between four and 24 hours post-infection. Steady-state patterns were non-gradient and genotype-specific, where mRNA levels from the G gene exceeded those from the more promoter-proximal nucleocapsid (N) gene across isolates. Transcript stabilities could not account for the non-gradient patterns observed, indicating that relative mRNA levels more strongly reflect transcription than decay. Our results indicate that gene expression from a small but diverse set of RSV genotypes is non-gradient and genotype-dependent. We propose novel models of RSV transcription that can account for non-gradient transcription.

**Competing interests:** The authors have declared that no competing interests exist.

## Introduction

Respiratory syncytial virus (RSV) can infect individuals repeatedly and is the most common pathogen associated with severe lower respiratory tract disease in children worldwide [1–5]. Numerous host-related and environmental risk factors for severe disease are known [6–8] while viral factors are less clear.

RSV is a nonsegmented negative-strand RNA virus (NSV) classified into two major subgroups, A and B, largely distinguished by antigenic differences in the attachment (G) protein [9, 10]. The two subgroups are estimated to have diverged from an ancestral strain over 300 years ago [11] and have evolved into multiple co-circulating genotypes [11–15].

Transcription in RSV and other NSV is sequential, with genes transcribed in their order of occurrence from the 3' promoter of the genome [16–22]. Each of the ten genes of RSV contains essential gene start (GS) and gene end (GE) signals flanking the open reading frame (ORF) [23–25]. Transcription is initiated at the GS signal which also serves as a capping signal on the 5' end of the nascent mRNA [21, 26, 27]. The polymerase then enters elongation mode until it reaches a GE signal, where the mRNA is polyadenylated and released [21, 23]. Two genes overlap at the 5' end of the RSV genome. The GE signal of matrix 2 (M2) occurs downstream of the GS signal of the large polymerase (L) gene. The polymerase must return from the M2 GE signal for full-length L mRNA to be made [28], suggesting that transcribing polymerases scan the RSV genome bidirectionally for a new GS signal after terminating transcription. Indeed, scanning polymerase dynamics may be a universal feature of NSV transcription [21, 29–31].

By homology with other NSV, it is widely assumed that transcription in RSV follows a gradient, where the extent to which a gene is transcribed falls with its distance from the 3' promoter [22, 32, 33]. This is believed to be due to transcriptional attenuation at the intergenic (IG) regions, although the mechanisms underlying the attenuation are unknown [34]. Earlier studies reported results consistent with a gradient [32, 35, 36]; however, recent studies show RSV mRNA abundances that peak at the G gene, which is the seventh gene downstream of the 3' promoter of the genome [33, 37]. We recently reported the G gene to be the most abundant in clinical samples obtained from RSV/A- and RSV/B-infected infants [38]. Thus, existing data suggest the possibility that RSV gene expression can be non-gradient and variable.

Here we explored the natural diversity of patterns of RSV gene expression by using qPCR to measure mRNA abundances of five different RSV genes [nonstructural genes one and two (NS1, NS2), nucleocapsid (N), attachment (G), and Fusion (F)] from isolates that we sequenced belonging to both subgroups and four genotypes (RSV/A/GA1, RSV/A/ON, RSV/B/GB1, RSV/B/BA). Genotype-dependent patterns were observed, all diverging from a gradient and all showing higher levels of G mRNA than expected. Transcript stabilities did not account for the non-gradient patterns, indicating that mRNA levels reflect transcription more than decay. We propose novel models incorporating the possible effects of stochastic transcription and/or the recycling of transcribing polymerases in order to begin rationalizing non-gradient RSV transcription.

## Results

### RSV mRNA abundances

Oligonucleotide standards of known concentration were used to convert cycle threshold ($C_T$) values measured by real-time PCR for mRNA targets (Fig 1A) to mRNA abundances. Twenty oligonucleotide standards and sets of reagents (primers and probe) (S1 Table) were needed to quantify 20 mRNA targets (five genes in four isolates). All reagents gave rise to a similar range of $C_T$ values for standards at equal concentration (Fig 1B).

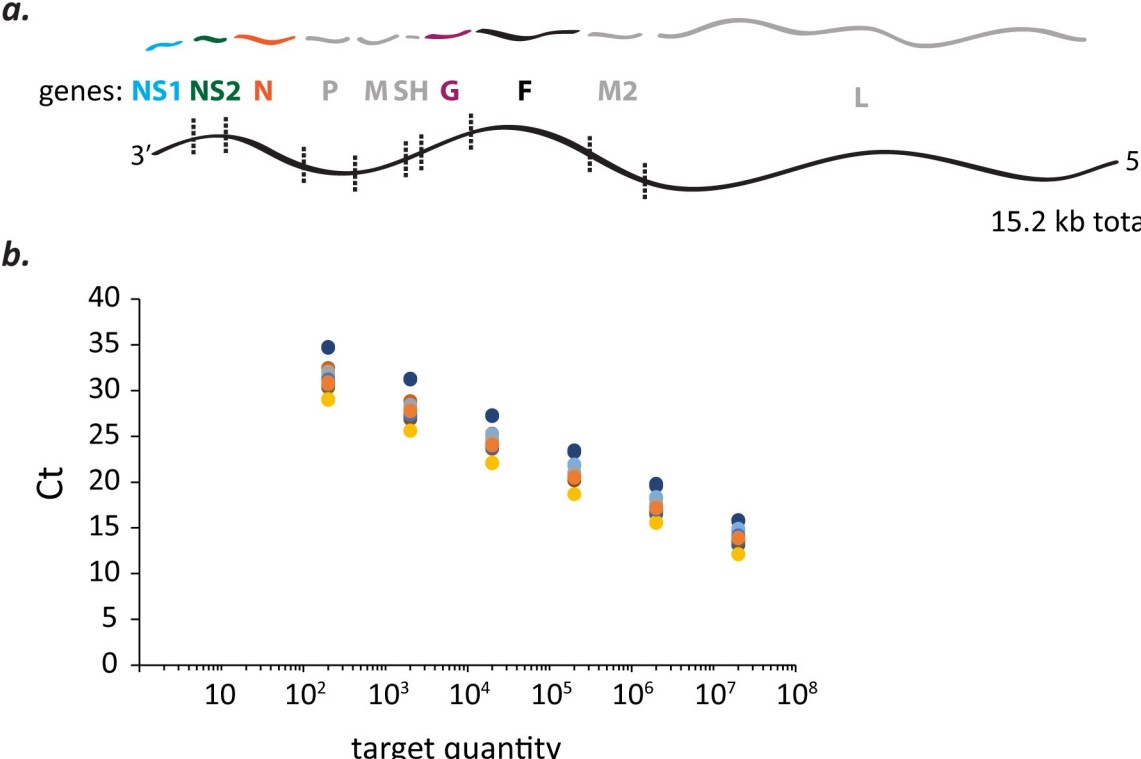

**Fig 1. qPCR-based measurements of mRNA abundances for five RSV genes from four isolates representing different genotypes using oligonucleotide standards. (a)** Five of 10 RSV genes were chosen for mRNA abundance measurements by qPCR. The 5 genes (NS1, NS2, N, G, & F) span half of the nucleotide length of the 15.2 kb genome and its entire gene length minus the final two genes, M2 and L. **(b)** Known amounts of different oligonucleotide standards were detected over a similar range of cycle threshold ($C_T$) values. Twenty different oligonucleotide standards at known concentrations were needed (4 virus isolates x 5 mRNA targets) to convert $C_T$ values measured for viral mRNA targets into mRNA abundances. Each dot represents the mean $C_T$ of duplicate measurements of an oligonucleotide standard at a known concentration or quantity (= number of molecules / qPCR rxn). Dots of like color are dilutions of the same oligonucleotide standard.

## Relative mRNA levels

RSV isolates from both major subgroups (A and B) and four genotypes (A/GA1/Tracy, A/ON/121301043A, B/GB1/18537, B/BA/80171) were used to infect HEp-2 cells (MOI = 0.01). Total mRNA abundances began to plateau at ~48 hours post-infection (pi) for all isolates (Fig 2A), consistent with the presence of significant viral cytopathic effect beyond this time-point. Relative mRNA levels were calculated for each isolate at each time-point by dividing the abundance of each mRNA by the total mRNA abundance (Fig 2B). Relative mRNA levels reached steady-state between four and 24 hours pi (Fig 2B).

Consistent with sequential transcription, the mean relative level of NS1 mRNA decreased for all isolates after four hours pi (S2 Fig). The significance of the drop was calculated using two regression models with either the relative level of NS1 mRNA as the dependent variable and time pi, four vs. greater than four hours, and genotype as the independent variables (p<0.001) or time pi as the dependent variable and the relative level of NS1 mRNA and genotype as the independent variables (p = 0.010).

All four sets of steady-state mRNA levels were non-gradient, with levels of G mRNA exceeding levels of N mRNA (Fig 3). Steady-state mRNA levels also showed both subgroup- and genotype-specific differences (Fig 3). Between subgroups, relative levels of NS1 and NS2 were most different (Fig 3), with the two being similar in RSV/A, and with NS1 levels

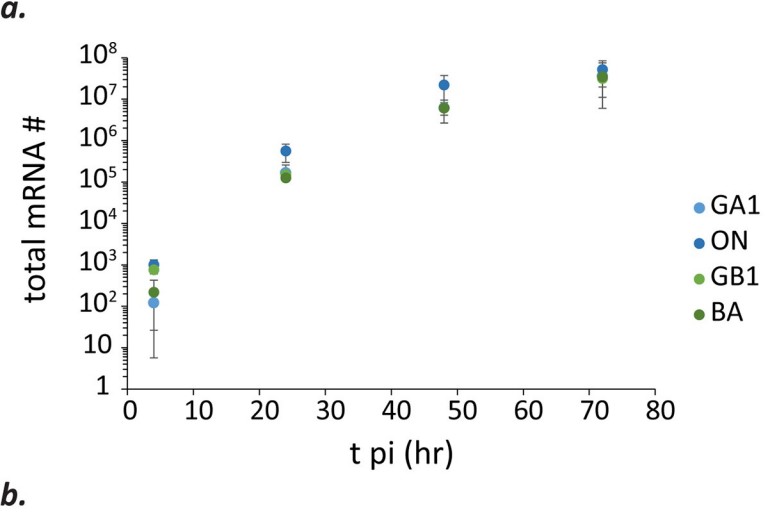

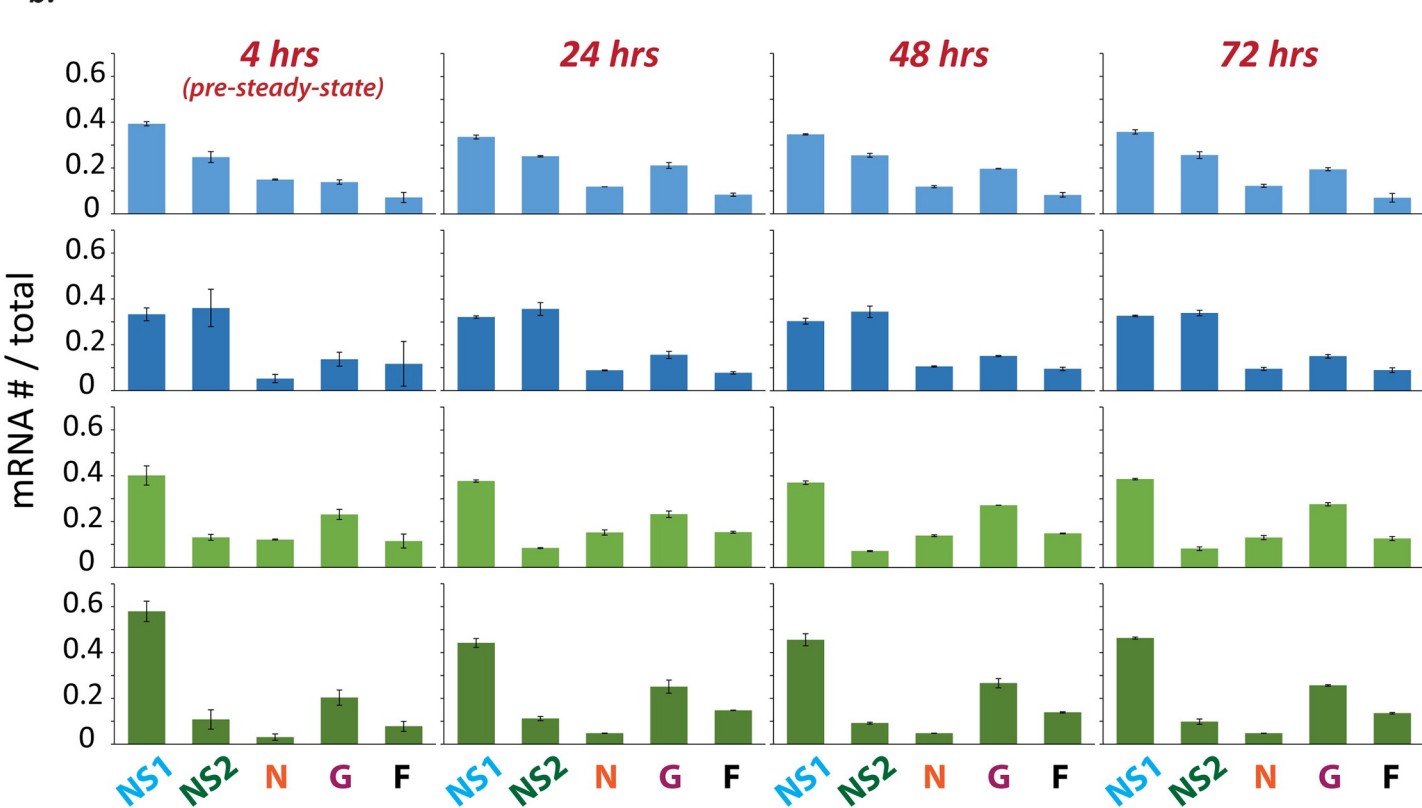

**Fig 2. Total mRNA abundances plateau beyond 48 hours post-infection and relative mRNA levels reach steady-state soon after the start of infection. (a)** Total mRNA abundances (= NS1+NS2+N+G+F) from HEp-2 cells infected with different isolates of RSV (MOI = 0.01) begin to plateau by ~48 hours post-infection (pi). Each dot (RSV/A/GA1Tracy [pale blue]; RSV/A/ON/121301043A [dark blue]; RSV/B/GB1/18537 [light green]; RSV/B/BA/80171 [dark green]) represents the mean and error bars the standard deviation of two independent experiments (n = 2). For each independent experiment, the mean was calculated from duplicate measurements and used in subsequent calculations. **(b)** Relative mRNA levels reach steady-state sometime between four and 24 hours pi. Histograms depicting relative mRNA levels are shown for all measured time-points (4, 24, 48, 72 hr pi) and all four isolates (color scheme same as (a)). Each bar depicts the mean mRNA # / total mRNA # of the indicated species and error bars show the standard deviation (n = 2). For each independent experiment, the mean was calculated from duplicate measurements and used in subsequent calculations.

exceeding NS2 by a factor of ~5 in RSV/B (Fig 3). Within RSV/A, the level of NS1 exceeded NS2 in the GA1 isolate, and was matched by NS2 in the ON isolate (Fig 3). In RSV/B, the level of G mRNA exceeded N in the BA isolate (~5-fold greater) more than it did in the GB1 isolate

(~2-fold greater) (Fig 3). Furthermore, genotype-specific steady-state mRNA levels were comparable in A549, Vero, and HEp2 cell lines (Fig 4A).

We explored whether relative mRNA levels might change in the context of a fully immuno-competent host. A pair of cotton rats was infected with each virus isolate and both lung lavage (LL) and nasal wash (NW) samples were collected at four days pi. Relative mRNA levels were genotype-specific and similar in cotton rat LL and NW samples, and comparable to those measured *in vitro* (Fig 4B).

### RSV mRNA stabilities and patterns of RSV gene expression

The observed divergence from a transcription gradient could be the result of differential stability of the RSV mRNAs. Therefore, we measured transcript stabilities by blocking transcription using the RSV RNA-dependent RNA polymerase (RdRp) inhibitor GS-5734 then monitoring mRNA levels by qPCR over time. Decay was measured for all five mRNAs from each of the four isolates in HEp-2 cells (Fig 5A). Exponential decay functions were fit to the data and half-lives were calculated from the decay constants. Half-lives ranged from 10 to 27 hours with a mean of $16 \pm 5$ hours (Fig 5B). Distributions of mRNA stabilities varied among the isolates, with GA1 having the greatest uniformity and lowest mean (= $12 \pm 1$ hours) (Fig 5A). Gene expression patterns were estimated by correcting measured mRNA abundances for degradation and recalculating relative mRNA levels (mRNA expressed = measured mRNA # * $e^{(\text{decay constant} * 24 \text{ hr})}$). Estimated levels of gene expression remained non-gradient; thus, differential mRNA stabilities do not account for the non-gradient patterns observed (Fig 5C). These data indicate that relative mRNA levels are 1) more strongly shaped by gene expression than decay and 2) can safely be interpreted to reflect levels of gene expression. The relative mRNA levels measured therefore constitute non-gradient and genotype-dependent patterns of RSV gene expression.

### Discussion

We observed non-gradient and genotype-dependent patterns of RSV transcription. A gene expression gradient has been widely assumed for RSV, but supporting data come from a

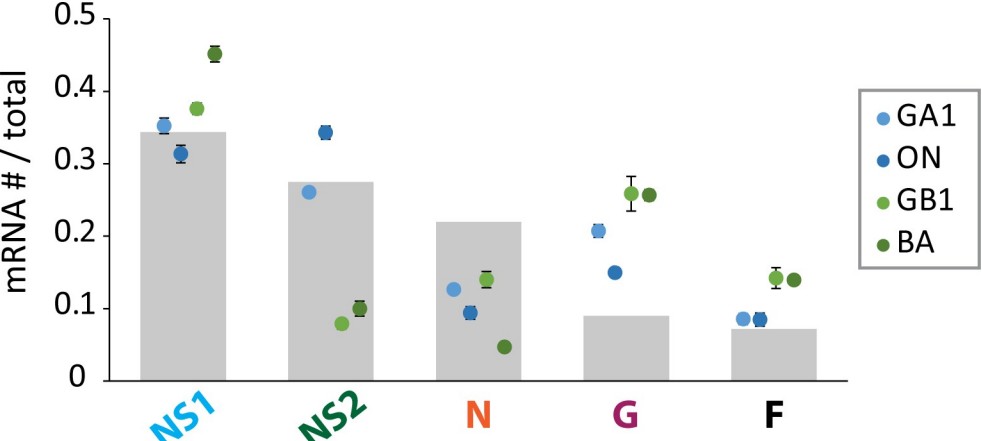

**Fig 3. Relative mRNA levels are genotype-specific and non-gradient.** Grey bars depict relative mRNA levels expected from an expression gradient resulting from a 20% decrease in transcription at every gene junction. Each dot depicts the mean mRNA # / total mRNA # observed for the indicated species and isolate (RSV/A/GA1Tracy [pale blue]; RSV/A/ON/121301043A [dark blue]; RSV/B/GB1/18537 [light green]; RSV/B/BA/80171 [dark green]) in HEp-2 cells (MOI = 0.01) at steady-state. Steady-state mean relative mRNA levels and standard deviation were calculated using the mean of each relevant time-point (24, 48, 72 hours post-infection). The mean of each time-point was calculated from two independent experiments, and the mean from each experiment was calculated from duplicate measurements as described.

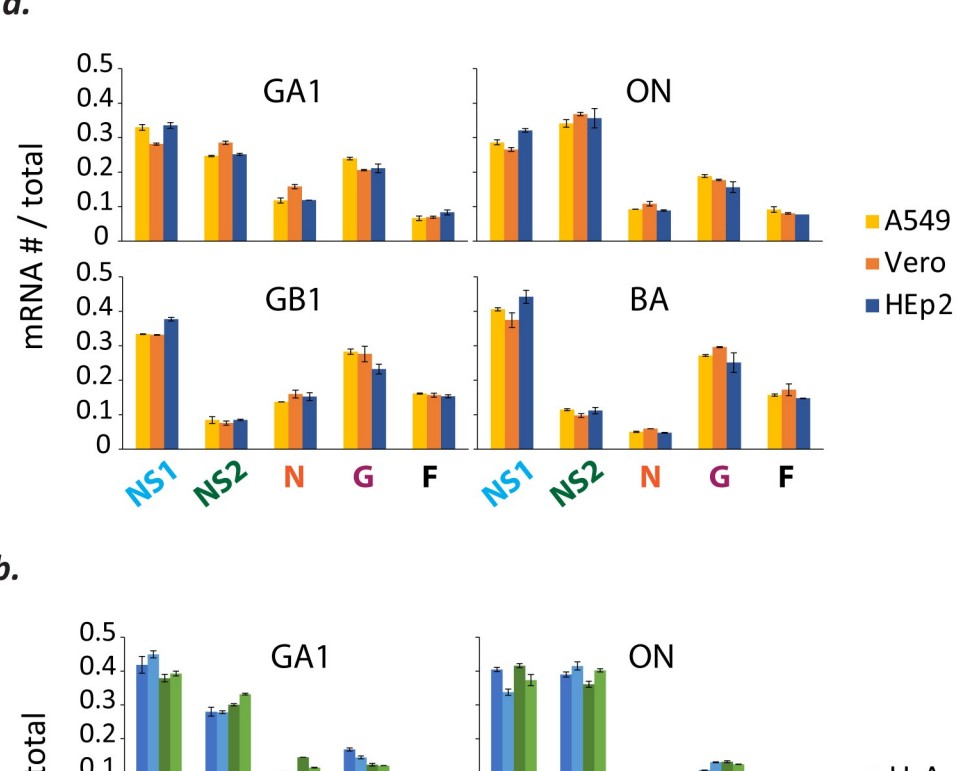

**Fig 4. Relative mRNA levels are comparable in different cell lines and in nasal wash and lung lavage samples from infected cotton rats. (a)** Relative mRNA levels are comparable in different cell lines. Viral mRNA levels were measured from infected A549 (in yellow), Vero (in orange), and HEp-2 (in blue) cell lines (MOI = 0.01) at 24 hours post-infection (pi). Each bar depicts the mean mRNA # / total mRNA # of the indicated species and error bars show the standard deviation (n = 2). For each independent experiment, the mean was calculated from duplicate measurements and used in subsequent calculations. **(b)** Relative mRNA levels are comparable in lung lavage (LL) and nasal wash (NW) samples from infected cotton rats. Each bar depicts the mean mRNA # / total mRNA # of the indicated species and error bars show the standard deviation calculated from duplicate measurements of the same sample. Results from LL samples collected 4 days pi are shown in blue (cotton rat A = light blue; cotton rat B = dark blue) and NW samples shown in green (cotton rat A = light green; cotton rat B = dark green).

modest number of studies and are largely restricted to laboratory-adapted isolates (Long and A2) from the prototypic GA1 genotype of subgroup A. The first measurements were made by Collins and Wertz (1983) using an A2 strain in HEp-2 cells [20, 36, 39]. They discovered the gene order of RSV and found it was approximated by decreasing mRNA abundances measured by northern blot [20, 36, 39]. Barik later reported a gradient by dot blot hybridization of radiolabeled mRNAs produced *in vitro* using ribonucleoprotein (RNP) complex from an RSV Long strain and cell extract from uninfected HEp-2 cells [35]. Over a decade later, Boukhvalova et al. measured a gradient-like pattern by qPCR of mRNA abundances from an RSV Long

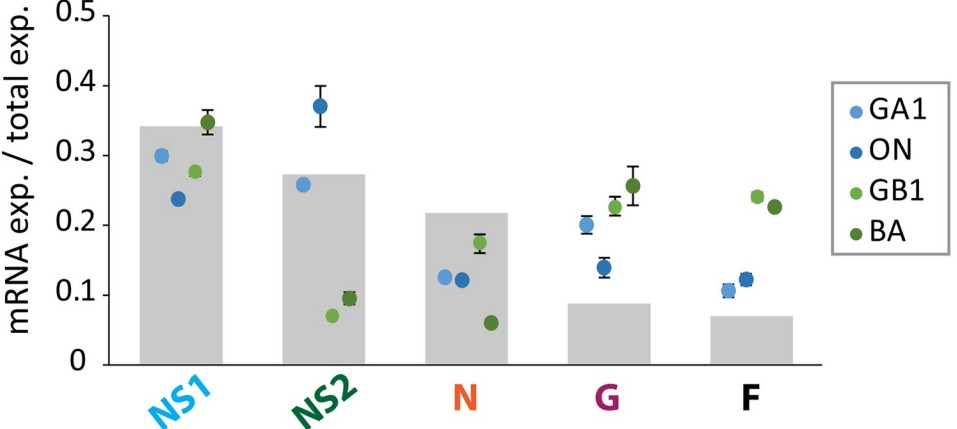

**Fig 5. Transcript stabilities do not account for non-gradient patterns, indicating that relative mRNA levels strongly reflect RSV gene expression. (a)** Viral mRNAs decay after addition of GS-5734, a viral polymerase inhibitor. Viral mRNA levels were divided by RNase P mRNA levels to control for well-to-well variation in the amount of sample obtained, then normalized. Each dot represents the mean normalized mRNA # and error bars the standard deviation of two independent experiments (n = 2). For each independent experiment, a mean was calculated from the means of two different samples; and each sample mean was obtained from duplicate measurements. **(b)** Decay constants obtained from exponential decay functions fit to each data set were used to calculate mRNA half-lives (RSV/A/GA1Tracy [pale blue]; RSV/A/ON/121301043A [dark blue]; RSV/B/GB1/18537 [light green]; RSV/B/BA/80171 [dark green]). **(c)** Transcript stabilities cannot account for non-gradient mRNA levels. Grey bars depict relative mRNA levels expected from an expression gradient resulting from a 20% decrease in transcription at every gene junction. Each dot depicts the mean expressed mRNA # / total expressed mRNA # estimated for the indicated mRNA species and virus isolate in HEp-2 cells (MOI = 0.01) at 24 hours post-infection (mRNA expressed = mRNA # observed * $e^{(\text{decay constant} * 24 \text{ hr})}$).

strain grown in A549 cells [32]. In contrast, Aljabr et al. recently reported mRNA abundances by RNA-Seq from an A2 strain in HEp-2 cells that are inconsistent with a gradient. The most abundant mRNA they observed was associated with the G gene [33]. Levitz et al. reported non-gradient mRNA levels and found the G gene to be the most highly expressed at later time-points in A549 cells infected with isolates from the RSV/B subgroup [37]. Thus, recent published data indicate that patterns of RSV gene expression vary and do not always follow a gradient. Here, we report data from isolates belonging to four different genotypes (GA1, ON, GB1, BA) and of variable passage number (GA1 and GB1 > 10, ON = 6, BA = 7) showing non-gradient and variable patterns of gene expression, and all with an apparent excess of G mRNA. These results require us to rethink existing models of RSV and NSV transcription.

Accurate mRNA abundance measurements by qPCR require reagents that bind target without any mismatches [40, 41]. Perfectly designed and distinct sets of reagents can amplify target with variable efficiency, as the amplification efficiency depends on the physicochemical properties of the reagents (the free energies of different intra- and intermolecular interactions) and the qPCR conditions used. For our 20 oligonucleotide standards, we found the lowest melting temperature from each set of reagents correlated positively with amplification efficiencies and negatively with cycle threshold values (S1 Fig). These correlations indicate that physicochemical differences in the primers and probes can account for the minor variation observed in the amplification of oligonucleotide standards, and support the accuracy of our approach to measuring viral mRNA abundances.

Among the genotype-dependent patterns of RSV transcription observed, the greatest difference occurred between subgroups A and B in the mRNA levels of NS1 and NS2. The similar levels of NS1 and NS2 from the RSV/A genotypes (GA1, ON) might partly be a result of frequent polymerase read-through from a weak NS1 GE signal [23]. Levels of NS2 are ~5-fold lower than NS1 from the RSV/B genotypes (GB1, BA), and these genotypes show conserved substitutions just outside of the canonical NS1 GE signal [23, 25]. It is possible that these substitutions promote more efficient termination of transcription at NS1, and, along with transcriptional attenuation at the NS1-NS2 junction, thereby cause less transcription of NS2. Regarding potential functional origins of the difference between A and B subgroups in the transcription of NS1 and NS2, it should be remembered that both G protein and NS2 can suppress interferon signaling [42, 43]. Perhaps the G protein of subgroup B is more active than that of subgroup A in suppressing the interferon response, relaxing the need for the higher level of NS2 transcription observed in the two RSV/A strains. If this is true, and assuming a lack of translational differences, then similar patterns of transcription should be observed for other A and B strains. The remaining differences among genotype-dependent transcription patterns likely result from more subtle genomic differences and differences in mRNA stabilities. It is also worth mentioning that patterns of RSV transcription show higher relative levels of NS1 and NS2 in cotton rat samples than samples from cell culture. This might reflect greater stringency on productive viral infection within a fully immunocompetent host.

Non-gradient gene expression requires some mechanism/s to alter the likelihood of transcription at different genes. To address this, we propose two basic and *a priori* mutually compatible models (Fig 6). Each model is biophysically reasonable and consistent with existing data. Our findings are relevant not only to understanding RSV but potentially also to the understanding of transcription and gene regulation in all other NSV.

RSV transcription is widely thought to be obligatorily sequential with attenuation at gene junctions (Fig 6A). Sequential transcription is well supported by existing molecular biological and ultraviolet (UV) transcriptional mapping data [16–18, 20], but we believe the idea of *obligatorily* sequential transcription should be reappraised. Obligatorily sequential transcription means that a polymerase must transcribe genes in their order of occurrence from the 3'

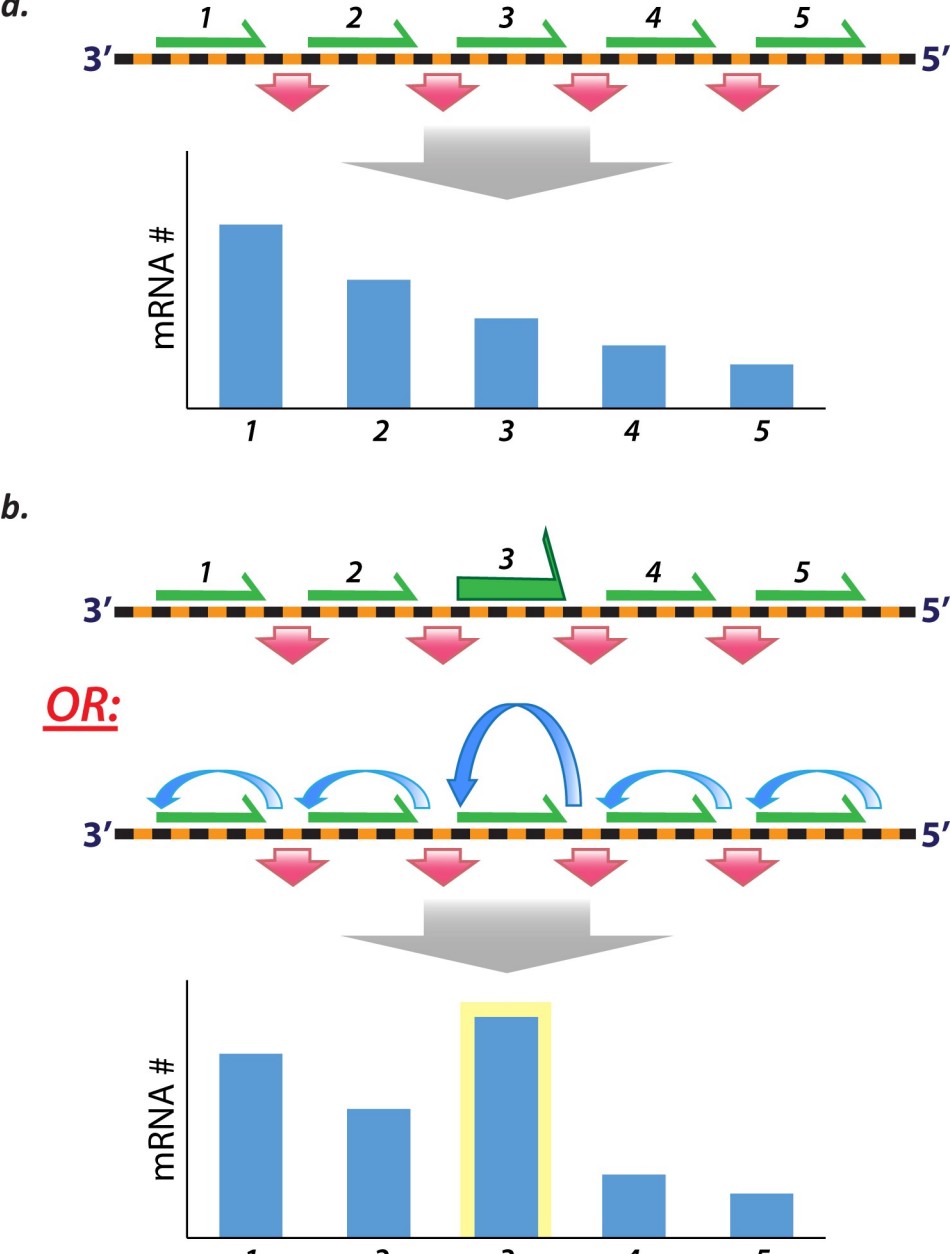

**Fig 6. Non-gradient transcription requires novel models of RSV transcription. (a)** The hitherto widely accepted model of RSV (and all NSV) transcription involves sequential transcription from the 3' promoter and transcriptional attenuation at gene junctions. Shown in black and orange is a cartoon representation of an idealized NSV genome. The green arrows represent equal probabilities of transcription at each of five genes, and the red arrows represent equal attenuation of transcription at each gene junction. Combined with sequential transcription, equal probabilities of transcription and equal rates of attenuation (and assuming equal transcript stabilities) result in a gradient of mRNA levels (shown as a plot depicting decreasing levels of mRNA from genes 1 to 5). **(b)** Non-gradient RSV transcription requires alternative models. Two such models incorporate either 1) transcription of variable probability (top cartoon–see green arrow of increased size indicating a higher probability of transcription at gene 3) and/or 2) polymerase recycling (bottom cartoon–see blue arrows indicating successive rounds of transcription of the same gene, with gene 3 supporting more rounds of transcription). Transcription of variable probability at different genes is supported by published data showing varied effects of different cis-acting sequences on gene expression. Polymerase recycling describes the repeated transcription of a single gene by one or more polymerases before proceeding to a downstream gene. Polymerase recycling would require polymerase scanning, which is well established, and potentially a mechanism to bias polymerase scanning away from the next downstream GS signal and back toward the upstream GS signal.

promoter; but, the initiation of transcription is a molecular event and is thereby stochastic. Thus, there is only ever a probability of transcription when a polymerase encounters a GS signal (Fig 6), and a polymerase can, in principle, scan past a GS signal before encountering another GS signal, downstream of the first, at which transcription is initiated. With sequential, but not obligatorily sequential, transcription, non-gradient gene expression is possible, and could result from higher probabilities of transcription initiation at one or more internal genes than those upstream (Fig 6B). Thus, the relative excess of G gene mRNA observed in our experiments could occur from polymerases, more often than not, failing to initiate transcription at the N gene before initiating at the G gene. Consistent with the idea of gene or locus-dependent rates of transcription, studies in RSV and other NSV show that gene expression depends on a variety of potentially mutable cis-acting factors [23–25, 44–46]. Moreover, polymerase recycling, which is not incompatible with stochastic and gene-dependent transcription, could also account for sequential and non-gradient transcription (Fig 6B). Polymerase recycling is the possibility of repeated rounds of transcription of a single gene by one or more polymerases. It fundamentally requires polymerase scanning and, in order to account for non-gradient transcription, some mechanism/s to bias, in a locus-dependent way, that scanning away from a downstream GS signal and toward the original upstream GS signal (Fig 6B). It is therefore possible that the N gene is usually expressed before the G gene, but G mRNA accumulates more because of polymerase scanning and increased re-initiation of transcription at the G gene GS signal. Either or both alternative models could help account for non-gradient transcription.

Our work establishes the existence of non-gradient and genotype-dependent transcription in RSV, and thus suggests the possibility of such gene expression in other NSV. We hypothesize that non-gradient transcription requires either 1) the initiation of transcription with unequal probabilities at different genes; and/or 2) locus-dependent polymerase recycling. Efforts should be made toward discovering and characterizing possible mechanisms, and integrating candidate mechanisms into simple, biophysically reasonable, and testable models of NSV transcription.

## Materials and methods

### Virus strains

RSV isolates were initially genotyped as described [13, 47] by sequencing a 270 bp fragment in the second hypervariable region of the G gene. RSV/A/GA1/Tracy and RSV/B/GB1/18537 are prototypic strains isolated in 1989 and 1962, respectively [13], while RSV/A/ON/121301043A and RSV/B/BA/80171 are contemporaneous strains isolated in 2013 and 2010, respectively [48, 49]. The viral pools used here of RSV/A/GA1/Tracy and RSV/B/GB1/18537 were passaged through in vitro cell culture > 10 times, the pool of RSV/A/ON/121301043A was passaged 6 times, and the pool of RSV/B/BA/80171 was passaged 7 times.

### Cell-lines and cotton rats

HEp-2 (ATCC CCL-23), A549 (ATCC CCL-185), and Vero (ATCC CCL-81) were cultured in minimal essential medium (MEM) containing 10% fetal bovine serum (FBS), 1 μg/ml penicillin, streptomycin, and amphotericin B (PSA), and supplemented with L-glutamine.

Male and female *Sigmodon hispidus* cotton rats were bred and housed in the vivarium in Baylor College of Medicine. Cotton rats were ~75 to 150 g of body weight at the start of the experiments.

## Viral replication in cell culture and cotton rats

The media from 70–90% confluent HEp-2, A549, or Vero cells in 24-well plates was aspirated, and 0.2 ml of virus diluted to a multiplicity of infection (MOI) of 0.01 in MEM containing 2% FBS with antibiotics, antifungal, and L-glutamine (2% FBS-MEM) was added to replicate wells for each of the time-points to be acquired. Plates were incubated at 37°C and 5% $CO_2$ for 1 hour. Following infection, virus-containing media was aspirated and replaced with 1 ml of pre-warmed 2% FBS-MEM. Plates were incubated at 37°C and 5% $CO_2$ until sample collection. At each time point, the media was aspirated and infected monolayers were lysed with 1X RIPA buffer and pelleted by centrifugation. The supernatant was flash frozen in a mixture of dry ice and 95% ethanol then stored at -80°C.

Eight- to ten-week-old male and female cotton rats were anesthetized with isoflurane gas and inoculated intranasally with $10^5$ plaque forming units (pfu) of RSV as described [50]. Cotton rats were euthanized with carbon dioxide on day 4 post-infection. Nasal wash (NW) samples were collected from each cotton rat by disarticulating the jaw and washing with 2 ml of collection media (= Iscove's media containing 15% glycerin and mixed 1:1 with 2% FBS-MEM) through each nare, collecting the wash from the posterior opening of the pallet. Lung lavage (LL) samples were collected after the left lung lobe was removed and rinsed in sterile water to remove external blood contamination and weighed. The left lobe was trans-pleurally lavaged using 3 mL of collection media. Both NW and LL fluids were stored at -80°C.

## RNA extraction and reverse transcription

Viral RNA was extracted from clarified cell lysates or samples obtained from cotton rats as described [48] by using the Mini Viral RNA Kit (Qiagen Sciences, Germantown, Maryland) and automated platform QIAcube (Qiagen, Hilden, Germany) according to the manufacturer's instructions. Complementary DNA (cDNA) was generated using the SuperScript™ IV First-Strand Synthesis System and oligo(dT)$_{20}$ primers according to the manufacturer's instructions (ThermoFisher Scientific).

## RSV mRNA abundance measurements

Accurate mRNA abundance measurements by qPCR require reagents that bind target without any mismatches [40, 41]. Twenty sets of target-specific primers and probes (from five mRNA targets for four virus isolates) were designed using whole genome sequences obtained by next-generation sequencing. $C_T$ values were measured using the StepOnePlus Real-Time PCR System (ThermoFisher Scientific). Thresholding was performed according to the manufacturer's instructions [51].

Oligonucleotide standards were used to convert sample $C_T$ values to mRNA abundances. Twenty oligonucleotide standards identical in sequence to the 20 targets of the specific primers and probes described above were purchased from IDT®, received lyophilized and resuspended in TE buffer pH 8. Each oligonucleotide standard was diluted to $4\times10^6$ molecules/μl and further diluted serially to a concentration of 40 molecules/μl in TE buffer. Duplicate $C_T$ values were measured for each dilution and an average $C_T$ was calculated. Average $C_T$ values and known amounts (molecules/rxn) were used to construct a standard curve for each oligonucleotide standard.

For cDNAs derived from *in vitro* cell lysate or cotton rat samples, $C_T$ values were measured in duplicate and used to calculate an average. Each average sample $C_T$ value was converted to an mRNA abundance using the linear relationship determined for the appropriate oligonucleotide standard $C_T$ vs. log10 of the oligonucleotide standard amount (molecules/rxn).

## RSV mRNA stability measurements

Samples of HEp-2 cells infected with virus isolates at an MOI of 0.01 were collected from single wells of 24-well plates at multiple time-points up to 48 hours after addition of 100 μM GS-5734. GS-5734 is a monophosphate prodrug of an adenosine nucleoside analog that binds a broad range of viral RNA-dependent RNA polymerases (RdRps) and acts as an RNA chain terminator [52, 53]. Samples were collected as described above using 1X RIPA buffer to lyse infected cells, clarifying the lysate by centrifugation, and flash-freezing and storing the clarified lysate at -80˚C. Viral RNA were extracted and converted to cDNA using oligo(dT)$_{20}$ primers. Transcript levels from RNase P (a host housekeeping gene) were measured using qPCR reagents acquired from the Centers for Disease Control and Prevention (CDC) and used to correct viral mRNA levels for well-to-well variation in the amount of sample obtained. Exponential decay functions were fit to the normalized data and used to calculate half-lives. Estimates of the amounts of mRNA expressed up to 24 hours pi were made by correcting the observed mRNA abundances at 24 hours pi for degradation using the exponential decay constants calculated (the number of expressed = the number of observed $^*\ e^{(\text{decay constant }^*\ 24\ \text{hr})}$) and assuming production of all observed mRNA at t = 0 hours post-infection. This unrealistic assumption maximizes the effect of different rates of decay on the estimated levels of total expressed mRNA.

## Whole genome sequencing and assembly

cDNAs for sequencing were generated from viral RNA using the SuperScript™ VILO™ cDNA Synthesis Kit and random hexamers (ThermoFisher Scientific). cDNAs were amplified using specific primers, and PCR products of each sample were purified and pooled [54]. Pooled PCR products (1 μg) were digested with the NEBNext dsDNA fragmentase kit (New England BioLabs, Inc., Ipswich, MA). Fragmented DNA was end-repaired with the NEBNext End Repair Module (New England BioLabs, Inc.). End-repaired DNA was ligated with the Ion P1 adaptor and unique Ion Xpress barcode adaptors (KAPA Adapter Kit 1–24; KAPABiosystems). Agencourt AMPure XP beads (Beckman Coulter, Inc., Brea, CA) were used to selectively capture DNA between 100 and 250 bp in length. All reaction products were purified with the Isolate II PCR kit (Bioline USA, Inc.). These libraries underwent nick translation and amplification. Experion Automated Electrophoresis System (Bio-Rad Laboratories, Inc., Hercules, CA) was used to confirm fragment lengths and molar concentrations. Equal molar amounts of all libraries were pooled and libraries were sequenced by Ion Proton™ System (ThermoFisher Scientific) generating 150 bp reads. Raw data, FASTQ and BAM files, were generated by the Torrent Suite™ Software (version 5.0.4; ThermoFisher Scientific).

Reads were assembled by Iterative Refinement Meta-Assembler (IRMA), which was designed for highly variable RNA viruses with more robust assembly and variant calling [55, 56]. IRMA v0.6.7 (https://wonder.cdc.gov/amd/flu/irma/) was used with an assembly module specifically designed for RSV.

## Regression analysis

R (R Core Team, 2018) was used to perform regression modelling in order to evaluate the significance of the observed decrease in the relative level of NS1 mRNA after four hours post-infection. Linear regression was used to model the relative level of NS1 mRNA treating isolate genotype and time post-infection (4 vs. > 4 hours) as independent factors. Using a generalized linear model, isolate genotype and the relative level of NS1 mRNA were used to model time post-infection (4 vs. > 4 hours). The relative level of NS1 mRNA was log-transformed in both instances to improve normality.

## Ethics statement

All experimental protocols were approved by the Baylor College of Medicine's Institutional Animal Care and Use Committee (IACUC) (license # AN-2307). All experiments were conducted in accordance with the Guide for Care and Use of Laboratory Animals of the National Institutes of Health, as well as local, state and federal laws.

## Accession numbers

Sequences reported in this study were deposited in GenBank database under accession numbers MG813977-MG813995.

## Supporting information

**S1 Fig. Amplification efficiencies positively correlate and $C_T$ values negatively correlate with the minimum melting temperature (min. $T_m$) of the target-specific qPCR reagents used. (a)** Pearson correlation for amplification efficiencies vs. min. Tm: $R = 0.57$, $p = 0.0086$. **(b)** Pearson correlations for $C_T$ values measured at the extremes of target quantity (200 and $2x10^7$ molecules / rxn) vs. min. Tm: $R = -0.65$, $p = 0.002$ and $R = -0.66$, $p = 0.0015$, respectively.
(TIF)

**S2 Fig. Relative levels of NS1 mRNA tend to decrease beyond four hours post-infection.** NS1 mRNA # / total vs. time post-infection (4 or > 4 hours). Each plotted point (RSV/A/ GA1Tracy [green triangle]; RSV/A/ON/121301043A [purple cross]; RSV/B/GB1/18537 [blue square]; RSV/B/BA/80171 [red box]) represents the mean from duplicate measurements of a single sample.
(TIF)

**S1 Table. Primer and probe sequences for qPCR-based measurements of RSV mRNA abundances.** Primer and probe sequences are shown 5' to 3'. All reagents were purchased from Integrated DNA Technologies (IDT®). All probes contained the same dye (5' 6-FAM) and quencher (3' ZEN).
(DOCX)

## Acknowledgments

Thanks to Michel Perron of Gilead for providing the viral RdRp inhibitor GS-5734 for use in experiments to measure RSV transcript stabilities. Thanks to Brian Gilbert from the Department of Molecular Virology and Microbiology at Baylor College of Medicine for providing the cotton rats needed to perform RSV infection studies. We thank Kim Tran and David Morgan Henke for technical assistance.

## Author Contributions

**Conceptualization:** Felipe-Andrés Piedra, Pedro A. Piedra.

**Data curation:** Felipe-Andrés Piedra, Do-Kyun Kim.

**Formal analysis:** Felipe-Andrés Piedra, Xueting Qiu, Do-Kyun Kim.

**Investigation:** Felipe-Andrés Piedra, Xueting Qiu, Michael N. Teng, Vasanthi Avadhanula, Annette A. Machado, Do-Kyun Kim, James Hixson, Justin Bahl, Pedro A. Piedra.

**Methodology:** Felipe-Andrés Piedra, Xueting Qiu, Michael N. Teng, Vasanthi Avadhanula, Do-Kyun Kim, James Hixson, Justin Bahl, Pedro A. Piedra.

**Project administration:** Felipe-Andrés Piedra, James Hixson, Justin Bahl, Pedro A. Piedra.

**Resources:** James Hixson, Justin Bahl, Pedro A. Piedra.

**Software:** Xueting Qiu, Justin Bahl.

**Supervision:** Michael N. Teng, Vasanthi Avadhanula, Annette A. Machado, James Hixson, Justin Bahl, Pedro A. Piedra.

**Validation:** Felipe-Andrés Piedra.

**Visualization:** Felipe-Andrés Piedra.

**Writing – original draft:** Felipe-Andrés Piedra.

**Writing – review & editing:** Felipe-Andrés Piedra, Xueting Qiu, Michael N. Teng, Vasanthi Avadhanula, Annette A. Machado, Do-Kyun Kim, Justin Bahl, Pedro A. Piedra.

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
