## [Decision Letter · Decision Letter 0]

22 Oct 2019

PONE-D-19-24823

Non-gradient and genotype-dependent patterns of RSV gene expression

PLOS ONE

Dear Dr. Piedra,

Thank you for submitting your manuscript to PLOS ONE. After careful consideration, we feel that it has merit but does not fully meet PLOS ONE’s publication criteria as it currently stands. Therefore, we invite you to submit a revised version of the manuscript that addresses the points raised during the review process.

Both reviewers raised a number of concerns that need to be addressed. Please address the concerns noted by both reviewers paying close attention to each of the concerns raised by Reviewer #1 and the final concern raised by Reviewer #2. In some cases additional experiments may need to be performed to fully address a reviewer concern.

We would appreciate receiving your revised manuscript by Dec 06 2019 11:59PM. To enhance the reproducibility of your results, we recommend that if applicable you deposit your laboratory protocols in protocols.io, where a protocol can be assigned its own identifier (DOI) such that it can be cited independently in the future. For instructions see: http://journals.plos.org/plosone/s/submission-guidelines#loc-laboratory-protocols

We look forward to receiving your revised manuscript.

Kind regards,

Steven M. Varga, Ph.D.

Academic Editor

PLOS ONE

Journal Requirements:

1.

Reviewers' comments:

Reviewer's Responses to Questions

**Comments to the Author**

1. Is the manuscript technically sound, and do the data support the conclusions?

Reviewer #1: Yes

Reviewer #2: Partly

2. Has the statistical analysis been performed appropriately and rigorously? 

Reviewer #1: Yes

Reviewer #2: I Don't Know

3. Have the authors made all data underlying the findings in their manuscript fully available?

Reviewer #1: Yes

Reviewer #2: Yes

4. Is the manuscript presented in an intelligible fashion and written in standard English?

Reviewer #1: Yes

Reviewer #2: Yes

5. Review Comments to the Author

Reviewer #1: Piedra et al make observations that challenge the widely held belief that a mRNA gradient exists in RSV due to obligatorily sequential transcription with attenuation at gene junctions. mRNAs of five genes are quantitated over time, both in vitro and in vivo, and stability of each mRNA is also examined. This is an interesting and well-written study with original data, which show that steady state mRNA levels are genotype-dependent and do not follow a gradient, and that the non-gradient cannot be attributed to variations in mRNA stability. The study has implications for other NSV as well, and is in reasonable agreement with recent findings from other groups using distinct techniques. The use of 4 virus variants that cover the A and B subgroups, and confirmation of the findings in multiple cell lines as well as in infected cotton rats, is a notable strength of this paper. Statistical significance is not addressed for most of the data, and more discussion on some of their findings will help the reader interpret the work.

Comments

- Conclusions are in general well supported but any indication of significance of observed differences is lacking. Authors claim for example that NS1 levels decrease for all isolates after four hours, but the mentioned differences seem small and it’s not clear if they are significant. Also, in Fig. 2 it is not clear whether the means represent the mean of two independent experiments or the mean of all samples over two independent experiments.

- Two major findings of mRNA quantitation (in addition to being non-gradient) are that G mRNA levels are relatively high and that relative NS2 mRNA levels vary considerably between A and B isolates; can the authors speculate on the meaning of their findings?

- In the B isolates, the G:NS2 mRNA ratio appears to differ between cell lines and cotton rats. Can the authors say something about this result?

- Lab-adaptation in previous studies is mentioned as one possible explanation for the difference in gradients or non-gradients measured between older and newer studies resp. Are the isolates used here not lab-adapted?

- Read-through transcription is briefly mentioned in the discussion in an attempt to explain differences in NS2 mRNA levels. Could complex read-through transcription play an important role in non-gradient steady state levels?

- NS2 mRNA was previously shown to have a short half-life. The current findings do not confirm the previous. Can the authors address this difference?

- The authors propose two models, variable probability of transcription and polymerase recycling. However, rather than each representing a model, variable probability is more like an outcome, for which polymerase recycling could be one of the potential mechanisms. There is no data that aims to elucidate or support a model by which the virus regulates probability of transcription.

Reviewer #2: In these studies the relative abundance of gene transcription is assessed across the RSV genome. Previous studies have suggested that the gene transcription is sequential whereas the authors of this manuscript provide preliminary data, verifying other data in the field, that there can be non-sequential gene transcription. The data are supportive of this concept and the discussion offers some suggested mechanisms of how this mechanism could occur. The order of gene transcription in different cell lines and in cotton rats are consistent and genotype specific. While interesting, there is no further exploration of the mechanisms that might differentially regulate these findings that is dependent upon genotype. Furthermore, and perhaps more interesting, no additional exploration of what might be the consequence to this mechanism in the different genotypes. Does the differential expression of different genes give an infective advantage? Perhaps it alters the host response to the virus in some manner? These data provide solid preliminary data for further, more indepth investigation as to the "biologic" function of such a mechanism and/or how this mechanism is facilitated/regulated by the different genotypes explored.

A final issue is whether these mechanisms develop due to the time passaged in tissue culture, with the more recently derived strains (although still several years) having a similar expression profile compared to the longer term strains, especially regarding the NS2 gene. The older, for example, have likely been differentially passaged many times in different long term cell lines, Hep2, Vero. This consideration is as likely to account for differences as any other pressure. The question may not be can it happen, rather does it normally happen and what impact does it have on the normal infectivity, success of the viral replication, and disease associated outcome in vivo.

6. PLOS authors have the option to publish the peer review history of their article (what does this mean?). If published, this will include your full peer review and any attached files.

Reviewer #1: No

Reviewer #2: No

---

## [Author Response · Author response to Decision Letter 0]

22 Nov 2019

November 21, 2019

Dear Dr. Vargas and Reviewers:

Thank you for reviewing our manuscript, ‘Non-gradient and genotype-dependent patterns of RSV gene expression.’ Thank you also for the opportunity to respond to the insightful comments made, and the important questions raised. We have done our best to answer fully. 

Please see the reviewers’ comments (in italics) and our responses (in bold) below.

Reviewer #1: 

Piedra et al make observations that challenge the widely held belief that a mRNA gradient exists in RSV due to obligatorily sequential transcription with attenuation at gene junctions. mRNAs of five genes are quantitated over time, both in vitro and in vivo, and stability of each mRNA is also examined. This is an interesting and well-written study with original data, which show that steady state mRNA levels are genotype-dependent and do not follow a gradient, and that the non-gradient cannot be attributed to variations in mRNA stability. The study has implications for other NSV as well, and is in reasonable agreement with recent findings from other groups using distinct techniques. The use of 4 virus variants that cover the A and B subgroups, and confirmation of the findings in multiple cell lines as well as in infected cotton rats, is a notable strength of this paper. Statistical significance is not addressed for most of the data, and more discussion on some of their findings will help the reader interpret the work.

Comments

- Conclusions are in general well supported but any indication of significance of observed differences is lacking. Authors claim for example that NS1 levels decrease for all isolates after four hours, but the mentioned differences seem small and it’s not clear if they are significant. Also, in Fig. 2 it is not clear whether the means represent the mean of two independent experiments or the mean of all samples over two independent experiments.

Two kinds of observed difference are central to the paper: 1) the difference between measured RSV mRNA abundances and those expected from a gradient; and 2) differences among measured steady-state gene expression patterns belonging to strains from four different genotypes of RSV. It is worth stating that both kinds of difference are clear from the plotted data (see Fig3), and it does seem that the reviewer agrees with this point. The decrease in relative levels of NS1 mRNA after 4 hours post-infection (pi) is slight but consistent across genotypes (avg. = 11±8%). We find this interesting because a decrease in transcription from more 3’ proximal genes is consistent with the change that would be expected between pre-steady-state and steady-state relative mRNA levels resulting from sequential transcription. In addition, and as mentioned in the text, decreases for GA1 and BA were larger (avg. = 17±7%), and these strains showed lower total amounts of mRNA at t=4 hr pi. A steeper drop and lower total mRNA at the start of the experiment suggest that our 4 hr measurements occurred closer to the start of transcription for GA1 and BA than ON and GB1. Finally, each bar in each histogram of Fig2 represents the mean of two independent experiments, and not the mean of all wells over two independent experiments. 

- Two major findings of mRNA quantitation (in addition to being non-gradient) are that G mRNA levels are relatively high and that relative NS2 mRNA levels vary considerably between A and B isolates; can the authors speculate on the meaning of their findings?

G protein is needed for both viral attachment to host cells and immunosuppression and evasion. Perhaps relatively high (and non-gradient) levels of G mRNA are required to make sufficient G protein (transmembrane and secreted) to support robust infection and transmission of virus. Also, both G protein and NS2 are involved in suppressing interferon signaling. It is possible that the G protein of subgroup B is more active than that of A in its NS2-like function, relaxing the need for the higher level of NS2 transcription seen here in the two strains belonging to the A subgroup. If this is true, and assuming a lack of translational differences, then similar patterns of transcription should be observed for other A and B strains. We have amended our discussion (lines 279-286) in order to incorporate our answer to this question. 

- In the B isolates, the G:NS2 mRNA ratio appears to differ between cell lines and cotton rats. Can the authors say something about this result?

This could be a result of different transcript stabilities; relative mRNA abundances for NS1 and NS2 are, regardless of genotype, greater in cotton rat (CR) samples than samples from in vitro cell culture. Furthermore, the F:NS2 mRNA ratio is the most different between cell lines and CRs, followed by the G:NS2. These differences might also reflect the impact of the innate and adaptive immune response of the cotton rat on the life cycle of the virus. 

- Lab-adaptation in previous studies is mentioned as one possible explanation for the difference in gradients or non-gradients measured between older and newer studies resp. Are the isolates used here not lab-adapted?

We mention in the discussion (lines 239-242) that most published data concerning RSV gene expression come from one of two lab-adapted strains (Long & A2) of subgroup A. This is stated more to indicate the dearth of viral diversity explored than to suggest a relationship between lab-adaptation and gradient gene expression. In fact, Aljabr et al. report non-gradient mRNA levels from an A2 strain. Our data do not support a relationship between gradient/non-gradient gene expression and more/less passage, and suggest that RSV gene expression depends more on subgroup and genotype. The two prototypic strains that we used (RSV/A/GA1/Tracy and RSV/B/GB1/18537) have been passaged more than ten times, while the two contemporaneous strains (RSV/A/ON/121301043A and RSV/B/BA/80171) have been passaged 6 and 7 times, respectively. Lines 256-257 (in discussion) and 362-365 (in materials and methods) have been added to make this clear. 

- Read-through transcription is briefly mentioned in the discussion in an attempt to explain differences in NS2 mRNA levels. Could complex read-through transcription play an important role in non-gradient steady state levels?

In the simplest case, read-through transcription should have the effect of making relative mRNA levels (from, for example, neighboring genes) less different not more, and should flatten 3’ to 5’ negative gradients and not produce ‘bumps’ in transcription; but the transcriptional ‘bumps’ leading to non-gradient steady-state mRNA levels could result from the coupling of read-through transcription and polymerase recycling. 

- NS2 mRNA was previously shown to have a short half-life. The current findings do not confirm the previous. Can the authors address this difference?

We are aware of published data (Evans JE, et al. Virus Res. 1996 Aug; 43(2):155-61) showing a short half-life for NS2 protein in BS-C-1 cells, but cannot come across data (outside of our own) for mRNA stability. Furthermore, Evans et al. showed the 40 kDa form of the NS2 protein to be stable while the 14.5 and 30 kDa forms rapidly disappeared during the pulse-chase experiments. 

- The authors propose two models, variable probability of transcription and polymerase recycling. However, rather than each representing a model, variable probability is more like an outcome, for which polymerase recycling could be one of the potential mechanisms. There is no data that aims to elucidate or support a model by which the virus regulates probability of transcription.

With all due respect, we prefer to separate these distinct, albeit not incompatible, possibilities. Here, variable probability of transcription refers to the local probability of transcription initiation – i.e., the probability of transcription initiation at a GS signal when a viral polymerase is near or ‘on’ it (see lines 322-327). At the very least this probability depends on the GS signal and neighboring (including intergenic) sequence (we have also obtained evidence suggesting that the alignment of a GS signal with nucleoprotein can alter its recognition by a polymerase (data not presented here)). Polymerase recycling over a certain gene more than others can increase transcription from that gene even if the local probability of transcription initiation is constant across GS signals. 

Reviewer #2: 

In these studies the relative abundance of gene transcription is assessed across the RSV genome. Previous studies have suggested that the gene transcription is sequential whereas the authors of this manuscript provide preliminary data, verifying other data in the field, that there can be non-sequential gene transcription. The data are supportive of this concept and the discussion offers some suggested mechanisms of how this mechanism could occur. The order of gene transcription in different cell lines and in cotton rats are consistent and genotype specific. While interesting, there is no further exploration of the mechanisms that might differentially regulate these findings that is dependent upon genotype. Furthermore, and perhaps more interesting, no additional exploration of what might be the consequence to this mechanism in the different genotypes. Does the differential expression of different genes give an infective advantage? Perhaps it alters the host response to the virus in some manner? These data provide solid preliminary data for further, more indepth investigation as to the "biologic" function of such a mechanism and/or how this mechanism is facilitated/regulated by the different genotypes explored.

We agree with the reviewer that our findings are solid and need to be studied in relation to their biological meaning within the human host. Such studies are currently ongoing but are not the topic of this report.

A final issue is whether these mechanisms develop due to the time passaged in tissue culture, with the more recently derived strains (although still several years) having a similar expression profile compared to the longer term strains, especially regarding the NS2 gene. The older, for example, have likely been differentially passaged many times in different long term cell lines, Hep2, Vero. This consideration is as likely to account for differences as any other pressure. The question may not be can it happen, rather does it normally happen and what impact does it have on the normal infectivity, success of the viral replication, and disease associated outcome in vivo.

Our data do not show a relationship between RSV gene expression and passage number, and suggest that RSV gene expression patterns depend more on subgroup and genotype. The two prototypic strains that we used (RSV/A/GA1/Tracy and RSV/B/GB1/18537) have been passaged more than ten times, while the two contemporaneous strains (RSV/A/ON/121301043A and RSV/B/BA/80171) have been passaged 6 and 7 times, respectively. Lines 256-257 (in discussion) and 362-365 (in materials and methods) have been added to make this clear. ____________________________________________________________________

We appreciate the chance to respond to the above comments and questions. We enthusiastically submit our revised manuscript and humbly await your decision. 

Sincerely,

Dr. Felipe-Andrés Piedra

Dr. Pedro A. Piedra

---

## [Decision Letter · Decision Letter 1]

9 Dec 2019

PONE-D-19-24823R1

Non-gradient and genotype-dependent patterns of RSV gene expression

PLOS ONE

Dear Dr. Piedra,

Thank you for submitting your manuscript to PLOS ONE. After careful consideration, we feel that it has merit but does not fully meet PLOS ONE’s publication criteria as it currently stands. Therefore, we invite you to submit a revised version of the manuscript that addresses the points raised during the review process.

Please address the remaining two concerns noted by reviewer #1.  No additional experiments should be necessary to address either of the concerns.

We would appreciate receiving your revised manuscript by Jan 23 2020 11:59PM. To enhance the reproducibility of your results, we recommend that if applicable you deposit your laboratory protocols in protocols.io, where a protocol can be assigned its own identifier (DOI) such that it can be cited independently in the future. For instructions see: http://journals.plos.org/plosone/s/submission-guidelines#loc-laboratory-protocols

We look forward to receiving your revised manuscript.

Kind regards,

Steven M. Varga, Ph.D.

Academic Editor

PLOS ONE

Reviewers' comments:

Reviewer's Responses to Questions

**Comments to the Author**

1. If the authors have adequately addressed your comments raised in a previous round of review and you feel that this manuscript is now acceptable for publication, you may indicate that here to bypass the “Comments to the Author” section, enter your conflict of interest statement in the “Confidential to Editor” section, and submit your "Accept" recommendation.

Reviewer #1: (No Response)

Reviewer #2: All comments have been addressed

2. Is the manuscript technically sound, and do the data support the conclusions?

Reviewer #1: Yes

Reviewer #2: Yes

3. Has the statistical analysis been performed appropriately and rigorously? 

Reviewer #1: No

Reviewer #2: N/A

4. Have the authors made all data underlying the findings in their manuscript fully available?

Reviewer #1: Yes

Reviewer #2: Yes

5. Is the manuscript presented in an intelligible fashion and written in standard English?

Reviewer #1: Yes

Reviewer #2: Yes

6. Review Comments to the Author

Reviewer #1: Piedra et al have adequately addressed most of the stated concerns. Questions about two of the concerns remain, see below.

1) Statistical significance.

As to the example in item 1), the mentioned decrease in NS1 levels after four hours is not clear in Fig. 2. No statistical significance has been added. The norm to determine and communicate the likelyhood that stated differences (whether appearing subtle or not) are real is to calculate p values.

2) mRNA ratio variation between cell culture and cotton rats.

* In answering this question, the authors argue that perhaps transcript stabilities cause this difference. However, a general point in the paper is that the non-gradient is not caused by variations in transcript stabilities. These seem to contradict?

* This difference between in vitro and in vivo stands out and is interesting and could be important in assessing the value of vitro differences. Therefore the authors should include this finding in the discussion.

Reviewer #2: The Authors have responded to all comments and have satisfactorily added discussion points and made changes to the manuscript.

7. PLOS authors have the option to publish the peer review history of their article (what does this mean?). If published, this will include your full peer review and any attached files.

Reviewer #1: No

Reviewer #2: No

---

## [Author Response · Author response to Decision Letter 1]

18 Dec 2019

December 18, 2019

Dear Dr. Vargas:

Please see the remaining two comments (in italics) and our responses (in bold) below.

Reviewer #1: Piedra et al have adequately addressed most of the stated concerns. Questions about two of the concerns remain, see below.

1) Statistical significance.

As to the example in item 1), the mentioned decrease in NS1 levels after four hours is not clear in Fig. 2. No statistical significance has been added. The norm to determine and communicate the likelyhood that stated differences (whether appearing subtle or not) are real is to calculate p values.

We have added a supplementary figure (S2Fig) depicting measured levels of NS1 at 4 hours post-infection (pi) and > 4 hours pi. The noted decrease is statistically significant when analyzed by regression modelling. The manuscript has been modified to incorporate this information (see lines 154-158 and 476-484). 

2) mRNA ratio variation between cell culture and cotton rats.

* In answering this question, the authors argue that perhaps transcript stabilities cause this difference. However, a general point in the paper is that the non-gradient is not caused by variations in transcript stabilities. These seem to contradict?

The data we present indicate that transcript stabilities cannot account for the non-gradientness of the mRNA levels measured, not that transcript stabilities cannot at all affect relative mRNA levels. Moreover, in vitro transcript stabilities were measured, not in vivo stabilities. We therefore do not know what effect (if any) in vivo transcript stabilities might have on the difference between RSV mRNA levels in samples from cell culture and cotton rats. 

* This difference between in vitro and in vivo stands out and is interesting and could be important in assessing the value of vitro differences. Therefore the authors should include this finding in the discussion.

We agree that the difference between in vitro and in vivo patterns of RSV transcription is interesting. We have added lines 292-95 to our discussion.

Thank you for the chance to respond to the above comments. We submit our revised manuscript and await your decision. 

Sincerely,

Dr. Felipe-Andrés Piedra

Dr. Pedro A. Piedra

---

## [Editor Report · Decision Letter 2]

23 Dec 2019

Non-gradient and genotype-dependent patterns of RSV gene expression

PONE-D-19-24823R2

Dear Dr. Piedra,

We are pleased to inform you that your manuscript has been judged scientifically suitable for publication and will be formally accepted for publication once it complies with all outstanding technical requirements.

With kind regards,

Steven M. Varga, Ph.D.

Academic Editor

PLOS ONE
---

## [Editor Report · Acceptance letter]

27 Dec 2019

PONE-D-19-24823R2 

Non-gradient and genotype-dependent patterns of RSV gene expression 

Dear Dr. Piedra:

I am pleased to inform you that your manuscript has been deemed suitable for publication in PLOS ONE. Congratulations! Your manuscript is now with our production department. 

With kind regards,

on behalf of

Dr. Steven M. Varga 

Academic Editor

PLOS ONE